# REPURPOSING FOUNDATION MODEL FOR GENERALIZABLE MEDICAL TIME SERIES CLASSIFICATION

## ABSTRACT

Medical time series (MedTS) classification is critical for a wide range of healthcare applications such as Alzheimer's Disease diagnosis. However, its real-world deployment is severely challenged by poor generalizability due to inter- and intra-dataset heterogeneity in MedTS, including variations in channel configurations, time series lengths, and diagnostic tasks. Here, we propose FORMED, a foundation classification model that leverages a pre-trained backbone and tackles these challenges through re-purposing. FORMED integrates the general representation learning enabled by the backbone foundation model and the medical domain knowledge gained on a curated cohort of MedTS datasets. FORMED can adapt seamlessly to unseen MedTS datasets, regardless of the number of channels, sample lengths, or medical tasks. Experimental results show that, without any task-specific adaptation, the repurposed FORMED achieves performance that is competitive with, and often superior to, 11 baseline models trained specifically for each dataset. Furthermore, FORMED can effectively adapt to entirely new, unseen datasets, with lightweight parameter updates, consistently outperforming baselines. Our results highlight FORMED as a versatile and scalable model for a wide range of MedTS classification tasks, positioning it as a strong foundation model for future research in MedTS analysis. *Code release upon acceptance.*

## 1 INTRODUCTION

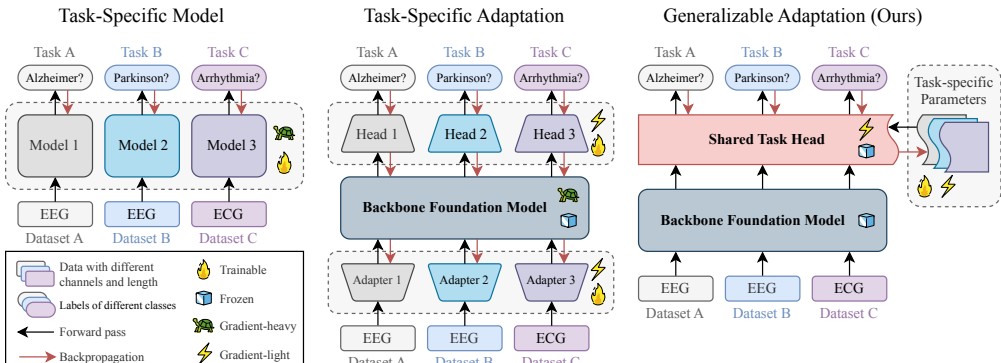

Figure 1: Paradigms of building models for different MedTS classification tasks. **Task-Specific Model (TSM):** Traditional classification models are designed for specific input shape and output classes, thus require retraining from scratch for each new dataset. **Task-Specific Adaptation (TSA):** By using a pre-trained and fixed backbone foundation models, the adaptation to new datasets requires training fewer parameters for each dataset, such as pre- and post-backbone adapters, which makes the combined model no longer applicable to other tasks, lacking generalization across tasks, and more prone to overfitting. **Generalizable Adaptation:** Generalizable adaptation is a post-backbone adaptation module that is shared across tasks of different datasets, which carries domain knowledge and transferable to unseen datasets with minimal training.

Medical time series (MedTS) classification, such as on electrocardiograms (ECG) and electroencephalograms (EEG), is critical for a wide range of medical scenarios such as diagnosing

Alzheimer's Disease (AD; Jeong (2004)), Parkinson's Disease (PD; Aljalal et al. (2022b;a)), and heart Arrhythmia (Jin et al., 2024b). Despite significant advancements in developing deep learning models for MedTS classification, several challenges still hinder their ability to generalize effectively across different datasets, and even among patients within the same dataset, posing a critical barrier to the successful translation of predictive algorithms into real-world clinical settings.

As MedTS data can be of various modalities and dimensions, in real-world scenarios, they often differ from the training data in all aspects. Therefore, developing machine learning models that can generalize across these diverse datasets is essential for translating predictive algorithms into clinical settings. However, there are three *unique* key challenges in MedTS that a generalizable model will have to face: (1) **Inter-dataset Heterogeneity**: The explicit characteristics of each dataset are diverse due to factors like the domain of physiological data, the equipment for data collection, etc. They require the model to be able to handle variations in the number of channels, sample duration, sampling rates, diagnostic targets, and so on (Ganapathy et al., 2018). (2) **Intra-dataset Heterogeneity**: Given the intractable nature of the underlying physiological state, even within the same dataset, heterogeneity still exists across the time of recording, experimental session, and most significantly, among patients due to the presence of noise and artifacts (Wang et al., 2024c; Ganapathy et al., 2018). As they are prevailing in the data, models are prone to overfitting to the training data, and thus show poor generalizability in real-world deployment. (3) **Data Insufficiency**: The other challenges could have been solved provided with sufficient data for deep learning methods, while in reality, available MedTS datasets are often small due to costly data collection or simply privacy concerns (Kaushik et al., 2020). Consequently, it increases the difficulty of training robust models that handle the above challenges effectively (Ganapathy et al., 2018).

To this end, developing a foundation model for MedTS classification requires capturing and sharing medical domain knowledge across tasks. Previous attempt such as Yang et al. (2023) adopts a Task-Specific Adaptation (TSA; see Figure 1) approach, in hope of capturing such knowledge in the backbone model. Yet the results of negative performance gain indicate that the model may only focus on extracting features that are informative for the training task, not of interest to future datasets, and thus lack generalization from task to task. Meanwhile, recent advances in foundation models bring sunlight for overcoming the above challenges by learning generic representations of time series data (Liang et al., 2024). However, they have focused predominantly on forecasting tasks (Ye et al., 2024; Wen et al., 2022), and simply applying TSA to adapt them for MedTS is not enough for capturing the sophisticated patterns for specific tasks from our preliminary results. Therefore, although they can serve as a great backbone model for extracting patterns in time series, dedicated effort in adaptation design is still required.

In this paper, we propose a novel approach to re-purpose foundation models pre-trained on large-scale, generic time series data for MedTS classification. We introduce FORMED, a **Fo**undation model **Re**purposed for **Med**ical time series classification, which achieves **generalizable adaptation** by seamlessly handling datasets with arbitrary channel configurations, dynamic time series lengths, and diverse diagnostic targets across multiple tasks (see Figure 1). Specifically, FORMED employs a pre-trained foundation model as a backbone, which captures general temporal features from time series data. We then adapt this backbone to the medical domain by integrating a specialized shell enriched with medical knowledge. This shell is trained on a curated cohort of MedTS data, enabling the model to effectively capture the unique characteristics of medical sequences. Even with far fewer samples in the MedTS cohort than during pre-training, FORMED retains strong generalization abilities and domain knowledge for MedTS classification tasks.

We curate a *repurposing cohort* of 5 MedTS datasets, including 2 ECG's and 3 EEG's, containing only 340K samples or 90 million time-points in total. These datasets feature diverse channel configurations (ranging from 12 to 33 channels), sample lengths (from 250 to 300 time-points), and diagnostic tasks (ranging from binary neurological to 5-class cardiovascular classification). Our evaluation focuses on two aspects: First, for datasets partially included in the repurposing cohort, FORMED achieves superior performance on unseen patients, outperforming 11 state-of-the-art TSA and TSM models across all five datasets. Second, for a completely new dataset not included in the cohort, FORMED can be efficiently adapted by updating only a small number of parameters while still achieving the highest accuracy and AUROC compared to TSM and TSA models, even across different data availability scenarios.

## 2 RELATED WORK

**Foundation Models for General Time Series**. To date, no foundation model has been specifically designed for time series classification tasks, let alone MedTS classification; instead, recent advances in time series foundation models mainly concentrate on forecasting tasks (Liang et al., 2024). Noticing their success in forecasting, it is as much tempting as theoretically and practically challenging to re-purpose these models for MedTS classification, yet these models all have major limitations such as design for *univariate time series* and requiring *Task-Specific Adaptations* (mid-column in Figure 1) that prevent them from being directly applicable to MedTS classification tasks (Cao et al., 2024; Sun et al., 2024; Chang et al., 2023).

For instance, Time-LLM (Jin et al., 2024a), UniTime (Liu et al., 2024) and GPT4TS (Zhou et al., 2023a) are all backboned on large language models, therefore they naturally handle time series data in a univariate manner, lacking the ability to integrate information across multiple channels for MedTS classification. Moreover, we empirically observe that time series foundation models that take LLM as backbone don't work well on time series datasets, in agree with Tan et al. (2024). Similarly, although TimeGPT (Garza et al., 2024) and TimesFM (Das et al., 2024) are pre-trained on large scale time series data, they treat co-evolving multivariate time series data as independent of each other, thus sharing the same limitation as the previous models. The one of its kind model, UniTS (Gao et al., 2024), is able to handle multivariate time series data and trained on multiple task domains including classification, yet due to its scale and design, it often requires fine-tuning of the whole model or performing prompt learning for optimal performance. This is both computationally expensive for gradients calculation (Figure 1), and more importantly, data-greedy due to the massive parameters to tune, making it not suitable for small-scale MedTS datasets.

Therefore, despite their effort and success, current foundation models require significant adaptation to meet the demands of MedTS classification effectively. This motivates the need for a specialized foundation model tailored to the complexities of MedTS, which can address these limitations with dedicated architectural components.

**Adaptation of Foundation Models for MedTS**. As general-purpose models, foundation models usually require various techniques to be effectively adapted for specific downstream tasks, including prompting, fine-tuning, re-programming and proposed re-purposing. Here we focus on re-programming and re-purposing as they can serve our purpose, see rest in Appendix A.

*Re-programming*: By reusing the pre-trained model's backbone (usually all the Transformer layers) without altering its internal weights, it leverages the model's existing capabilities. It is able to handle new domain of data or type of task, by wrapping the backbone model with input adapters and task heads (Figure 1 TSA). Yet on its dark side, the re-programmed model no longer serves as a general-purpose model, as both the input adapters and the task heads are task-specific, making it incapable of generalizing across tasks and datasets (Tan et al., 2024).

*Re-purposing*: Proposed in this work, it focuses on adapting the model to a new type of task with minimal modification to the task head only and refrains from being specific to certain task. Therefore, the repurposed model remains a general-purpose model for the field, which can serve as a new foundation model and be further adapted to new datasets efficiently (see Section 3). Its generalizability, data-efficiency and domain-expert nature make it extremely suitable for adapting time series foundation models for MedTS classification tasks.

**Forecasting v.s. Classification**. Although both forecasting and classification are key tasks in time series analysis, they are fundamentally different in nature. The primary distinction lies in the relationship between the input and output spaces. In forecasting, the model predicts future values within the same domain as the input, *i.e.*, mapping sequence → sequence (Lim & Zohren, 2021). For example, using past EEG signals to predict future EEG signals (Wang et al., 2024a). In contrast, classification uses the input to predict a categorical label, *i.e.*, sequence → category (Ali et al., 2019), such as diagnosing a neurological disease from EEG data (Wang et al., 2024a). In essence, forecasting only involves mapping input to output within the same domain, whereas in classification, the mapping is from one domain to a variety of others. Therefore, *adapting a forecasting model for general classification tasks requires more than simply modifying the prediction layer; it demands a comprehensive redesign and a deeper understanding of the problem space.*

## 3 PROBLEM STATEMENT

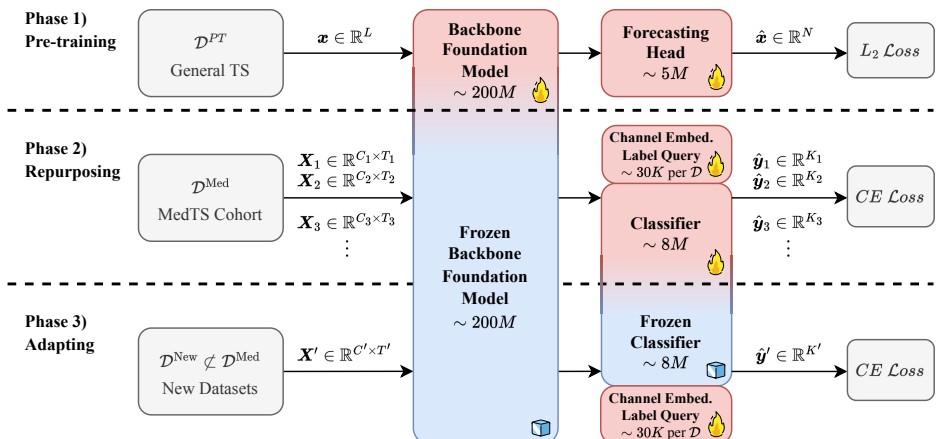

Figure 2: The three-stage process of adapting a time series foundation model for MedTS classification tasks. 1) **Pre-training** is already done on a cohort of diverse general time series datasets with forecasting tasks. 2) **Repurposing** the foundation model involves changing the forecasting head to a classification head, while keeping the rest of the model fixed, and the new model is then trained on a cohort of MedTS datasets to capture domain knowledge in MedTS. 3) **Adapting** the repurposed model to the new MedTS datasets with minimal training, where few parameters are adjusted for the new dataset and task, while the majority of the model remains fixed. The backbone foundation model is frozen in pre-training while trainable in repurposing and adapting. The classifier is trainable in repurposing while frozen in adapting.

Foundation models have showcased their capability in capturing general time series patterns, through pre-training on forecasting tasks, yet modifying them into foundation classification model for MedTS is not as straightforward. Here we define the problem and the key concepts involved in the adaptation process from a forecasting model into a general-purpose classification model.

**Definition 1** *Repurposing: The process of changing the objective of a pre-trained foundation model to a type of tasks that it was not originally trained for, by replacing and training a relative small output network while keeping the majority of the model fixed.*

The original pre-trained model contains a backbone model $f$ for representation learning and a forecasting head $g$ that predicts the horizon from the representations. It takes the input $\boldsymbol{x} \in \mathbb{R}^L$ from last $L$ steps of a univariate time series and predicts horizon $\hat{\boldsymbol{x}} \in \mathbb{R}^N$ in the next $N$ steps, which is essentially a dynamic mapping as $L$ and $N$ can vary:

$$g \circ f : \mathbb{R}^L \to \mathbb{R}^N \tag{1}$$

We replace the forecasting task head $f$ with a classification task head $h$, forming a new model that takes extra parameters $\boldsymbol{E} \in \mathbb{R}^{C \times D}$ and $\boldsymbol{Q} \in \mathbb{R}^{K \times D}$ for indicating the task-specific channels and classes, respectively, with $D$ as the model dimension, $C$ as the number of channels, and $K$ as the number of classes. The new model is then trained on a curated list of MedTS datasets $\mathcal{D}^{\mathrm{Med}}$, where the input $\boldsymbol{X} \in \mathbb{R}^{C \times T}$ is a multivariate time series data with $C$ channels and $T$ time steps, and the output $\hat{\boldsymbol{y}} \in \Delta^K$ is a label prediction where $\Delta^K = \left\{ \boldsymbol{d} \in [0,1]^K : \sum_{i=1}^K d_i = 1 \right\}$ is the probability simplex for $K$ classes. This is also a dynamic mapping as $C$, $T$ and $K$ may vary across datasets:

$$h \circ f : \mathbb{R}^{C \times T} \times \mathbb{R}^{C \times D} \times \mathbb{R}^{K \times D} \to \Delta^K \tag{2}$$

**Definition 2** *Adapting: The process of adjusting a repurposed foundation model to new datasets and tasks with a few data- or task-related parameters, while keeping the majority of the model fixed.*

After repurposing, the new model $h \circ f$ captures all the domain-specific knowledge in MedTS classification tasks, and can handle varying number of channels, length of input and number of classes

(Section 4), therefore it is fixed for new coming datasets and tasks. For new datasets $\mathcal{D}^{\text{New}}$ with $C'$ channels, $T'$ time steps, and $K'$ classes, the model is adapted to the new dataset by constructing the $\boldsymbol{E}' \in \mathbb{R}^{C' \times D}$ and $\boldsymbol{Q}' \in \mathbb{R}^{K' \times D}$, which is trained by calculating the loss $\mathcal{L}(\hat{\boldsymbol{y}}, \boldsymbol{y})$ on the new dataset:

$$(\boldsymbol{E}', \boldsymbol{Q}') = \underset{\boldsymbol{E}', \boldsymbol{Q}'}{\arg\min} \sum_{(\boldsymbol{X}', \boldsymbol{y}') \in \mathcal{D}^{\text{New}}} \mathcal{L}\left((h \circ f)(\boldsymbol{X}', \boldsymbol{E}', \boldsymbol{Q}'), \boldsymbol{y}'\right) \tag{3}$$

# 4 MODEL ARCHITECTURE

## 4.1 FEATURE EXTRACTOR FOR VARIABLE LENGTH TIME SERIES

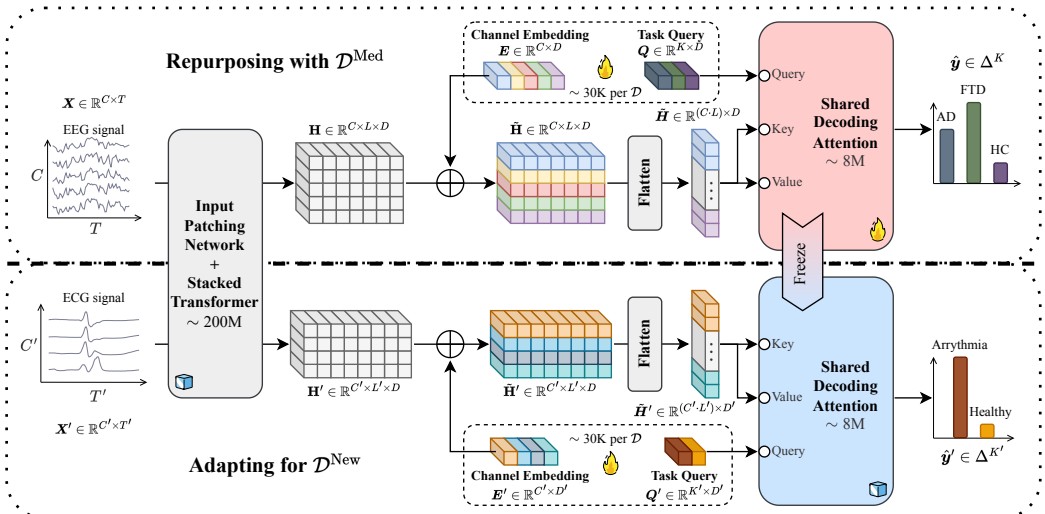

Figure 3: The architecture of the proposed model in repurposing and adapting. The backbone foundation model acts as a feature extractor and remains frozen all the time. The **Channel Embeddings** (CEs) and **Label Queries** (LQs) are task-specific parameters that are learned during both repurposing and adapting, and new ones will be created and learned if encountering new datasets. The **Shared Decoding Attention** (SDA) is a shared Transformer decoder layer that captures the interaction between all the features and classes, which once get trained on curated MedTS datasets $\mathcal{D}^{\text{Med}}$ during repurposing, will be fixed and reused when adapting to all future datasets and tasks $\mathcal{D}^{\text{New}}$. The $\oplus$ denotes broadcast addition.

We take TimesFM (Das et al., 2024) as the backbone for repurposing based on our preliminary comparative analysis of existing time series foundation models. TimesFM is pre-trained on a largest-scale dataset of diverse time series data for forecasting tasks and is able to capture general time series patterns within dynamic length of historical input. To repurpose it for MedTS classification, we can break down the model's anatomy into three parts, the input patching network, the stacked Transformer, and the output prediction network.

**Input Patching Network.** Given a univariate time series input $\boldsymbol{x} \in \mathbb{R}^T$ and binary mask $\boldsymbol{m} \in \{0,1\}^T$ with length $T$, they are first broken up into patches $\boldsymbol{X} \in \mathbb{R}^{L \times P}$ and $\boldsymbol{M} \in \{0,1\}^{L \times P}$ in a non-overlapping fashion, where $P$ is the patch size and $L = \lceil \frac{T}{P} \rceil$ is the number of tokens. Each patch $\boldsymbol{X}_{i,:}$ is the concatenation of $P$ consecutive elements of the input sequence $\boldsymbol{x}$ in a non-overlapping fashion and so is the $\boldsymbol{M}_{i,:}$. The $\boldsymbol{X}_{i,:}$ and $\boldsymbol{M}_{i,:}$ denote the $i$-th row of $\boldsymbol{X}$ and $\boldsymbol{M}$, respectively. The sequence of patches $\boldsymbol{X}$ and $\boldsymbol{M}$ are then projected to a sequence of tokens $\boldsymbol{Z} \in \mathbb{R}^{L \times D}$ in the model dimension $D$ using an input residual block:

$$\boldsymbol{Z}_{i,:} = \texttt{InputResidualBlock}(\boldsymbol{X}_{i,:}; \boldsymbol{M}_{i,:}) \tag{4}$$

**Stacked Transformer.** Before passing into the stacked Transformer, the positional encoding will be added to the tokens to form the input sequence $\tilde{\boldsymbol{Z}} \in \mathbb{R}^{L \times D}$. The stacked Transformer is then applied to the input sequence $\tilde{\boldsymbol{Z}}$ to capture the temporal dependencies and extract features using

casual self-attention, outputting feature rich tokens $\boldsymbol{H} \in \mathbb{R}^{L \times D}$:

$$
\begin{aligned}
\tilde{\boldsymbol{Z}}_{i,:} =& \boldsymbol{Z}_{i,:} \oplus \texttt{PositionalEncoding}(i) \\
\boldsymbol{H}_{i,:} =& \texttt{StackedTransformer}(\tilde{\boldsymbol{Z}}_{1,:}, \tilde{\boldsymbol{Z}}_{2,:}, ..., \tilde{\boldsymbol{Z}}_{i,:}; \dot{m}_1, \dot{m}_2, ..., \dot{m}_i)
\end{aligned}
\tag{5}
$$

where $\dot{m}_i = \min\{\boldsymbol{M}_{i,:}\}$ is the mask for the $i$-th patch for masking out completely empty ones.

**Output Prediction Network.** The output prediction network is a residual block layer that maps the last output $\boldsymbol{H}_{L,:}$ from the Transformer back to the original input spaces $\hat{\boldsymbol{x}} \in \mathbb{R}^N$, forming the prediction of the next $N$ time steps:

$$
\hat{\boldsymbol{x}} = \texttt{OutputResidualBlock}(\boldsymbol{H}_{L,:})
\tag{6}
$$

In summary, the duty of prediction lies solely on the last output prediction network, while the input patching network plus the stacked Transformer can be viewed as a feature extractor that maps the input time series $\boldsymbol{x}$ to a sequence of feature tokens $\boldsymbol{H}$ (Figure 3). This can be easily extended to process multivariate MedTS by processing each channel of input individually and stack the extracted features as $\mathsf{H} \in \mathbb{R}^{C \times L \times D}$ for data of $C$ channels. This will serve as the backbone feature extractor for the downstream classification model.

## 4.2 ATTENTION-BASED CLASSIFIER FOR INCONSTANT CHANNEL AND CLASS

Instead of using a simple linear classifier, as other time series classification models (Zerveas et al., 2021; Yang et al., 2023), which will require the input, output or both to have fixed number of channels and classes, we propose to use a Transformer decoder layer (Vaswani et al., 2017) for tackling such variability. Although inspired by techniques commonly employed in object detection (Carion et al., 2020) and image classification (Meng et al., 2023), it introduces key modifications tailored to address the *unique challenges* in MedTS classification. Our design is optimized for practical use, with an aim at handling dynamic shape input and outputing dynamic number of output classes, while reducing the computational overhead for training on new dataset and lowering risk of overfitting. Our attention-based classifier contains three key components: Channel Embeddings (CEs), Label Queries (LQs) and Shared Decoding Attention (SDA).

**Channel Embeddings.** As MedTS often exist in a multi-variate manner, injecting information about the channel will help the classifier distinguish between channels, thus promoting a more robust correspondence between the task and specific channel features. The Channel Embeddings are lightweight parameters that are grouped into a look-up table that maps the name of dataset to learnable channel embeddings $\boldsymbol{E} \in \mathbb{R}^{C \times D}$. These embeddings are then added to the feature tokens $\mathsf{H}$ to form the prompted feature tokens $\tilde{\mathsf{H}} \in \mathbb{R}^{C \times L \times D}$:

$$
\tilde{\mathsf{H}}_{:,i,:} = \mathsf{H}_{:,i,:} \oplus \boldsymbol{E}
\tag{7}
$$

**Label Queries.** Just as CEs, the label queries $\boldsymbol{Q} \in \mathbb{R}^{K \times D}$ are also task-specific, learnable embeddings, where $K$ is the number of classes for the given task. These task-specific queries are used to guide the attention mechanism to focus on the relevant features for the specific task. The label queries independently attend to the prompted feature tokens $\mathsf{H}'$ to find evidence for each class.

**Shared Decoding Attention.** The core evidence-finding process in classification is achieved through a shared decoding attention mechanism. It is a single decoder layer similar in Vaswani et al. (2017), that performs multi-head attention using $\boldsymbol{Q}$ as queries and $\tilde{H}$ as keys and values, where $\tilde{H} = \texttt{Flatten}(\tilde{\mathsf{H}}) \in \mathbb{R}^{(C \cdot L) \times D}$. It is followed by a residual block to obtain the logits $\hat{\boldsymbol{y}} \in \mathbb{R}^K$ for each class, where the probability prediction can be obtained using softmax or sigmoid functions depending on the type of task:

$$
\hat{\boldsymbol{y}} = \texttt{ResidualBlock}\left(\texttt{MultiHeadAttention}(\boldsymbol{Q}, \tilde{\boldsymbol{H}}, \tilde{\boldsymbol{H}})\right)
\tag{8}
$$

Note that all the parameters in SDA is independent on either input length, number of channels, or number of classes, therefore it is able to handle the inconstant channel, length and class in MedTS classification tasks. Moreover, as it defines how the task queries will interact with the prompted feature tokens and is shared across datasets and tasks, it is coerced to learn a shared dynamics and form a domain knowledge that is fixed and can be reused in adapting to new classification tasks.

### 4.3 Repurposing and Adapting

During repurposing, the backbone foundation model is frozen, and the weights in SDA is randomly initialized. For each dataset in our MedTS cohort, a pair of $E$ and $Q$ is also randomly initialized. These are then trained over the MedTS cohort to update their parameters (see Repurposing in Figure 2). After repurposing, the SDA should already capture the domain knowledge required for MedTS classification, thus it will be fixed and reused when adapting to new datasets and tasks. For unseen datasets that need to be classified, a new pair of $E$ and $Q$ will be created and learned during adapting, while the majority of the model remains fixed (see Adapting in Figure 2).

**Summary.** Our approach enables generalizability to new datasets, making it particularly suited for MedTS classification, and serves as a strong foundation model for all future MedTS tasks. In particular, our design brings significant benefits in overcoming the aforementioned challenges:

- **Generalizability Across Datasets:** The backbone foundation model is able to capture general time series patterns and is fixed for all datasets, while the SDA is independent of channel number, input length or class number, so that it can ben shared across datasets, and gains domain knowledge during repurposing. This ensures that the model never overlooks general patterns in the data, and also gains sufficient domain knowledge for MedTS classification. FORMED can be effectively generalized to datasets with different sample length, channels, and classes.
- **Generalization Across Subjects Within Dataset:** The $E$ and $Q$ are the only task-specific parameters that are dependent on the dataset and task, and they are used to guide the model to focus on the relevant features for the specific task. As their number of parameters is very limited, it is highly unlikely to memorize the specific pattern of the training data, keeping the model highly performant across diverse patients.
- **Lowered Data Requirement:** As the SDA is shared in all tasks, it can be trained on a joint of diverse small MedTS datasets as a whole, without the need for a single, large and comprehensive dataset which doesn't exist in practice. On the other hand, as the majority of the model parameters is fixed during adapting, the task-specific $E$ and $Q$ can be easily tuned with a little data from the new dataset. This design significantly reduces the data requirement for repurposing and adapting, making it particularly suitable for MedTS classification tasks with limited data.

## 5 Experiments

**Datasets.** We select 5 MedTS datasets to formulate a MedTS cohort and use it for repurposing (Figure 2). These datasets provide a broad range of physiological signals, capturing both cardiac and neurological activity, which are among the most commonly analyzed modalities in MedTS. See Table 3 for details on the datasets. Moreover, we also include an unseen, out-of-domain dataset (Liu et al., 2016) to assess our model's ability to generalize to new tasks. All datasets are split into train-test-valid sets following the patient-independent setting as in Wang et al. (2024c). The datasets span a wide range of channels, sampling rates, sample durations, and disease labels, allowing for the evaluation of inter-dataset heterogeneity. We use inter-subject variation, a key contributor to intra-dataset heterogeneity (Wang et al., 2023), as a proxy to assess the generalization capability of FORMED.

**Baselines.** We compare FORMED with 11 SOTA baselines including 10 TSM and 1 TSA models. The TSM models, including Autoformer (Wu et al., 2021), Crossformer (Zhang & Yan, 2022), FEDformer (Zhou et al., 2022b), Informer (Zhou et al., 2021), iTransformer (Liu et al., 2023), MTST (Zhang et al., 2024), Nonformer (Liu et al., 2022), PatchTST (Nie et al., 2022), Reformer (Kitaev et al., 2020) and Transformer (Vaswani et al., 2017), are included for comparing our model's performance on seen tasks to verify the applicability of repurposing. The additional TSA model, *PatchTST-TSA*, is modified from PatchTST by adding task-specific classification heads on top of the backbone model and trained on all datasets jointly from scratch. Due to its architectural similarity to TimesFM (Das et al., 2024), we use it to evaluate both the quality of repurposing and adapting.

**Evaluations.** The effectiveness of our method is demonstrated through the performance in terms of accuracy, precision, recall, F1 score, AUROC, and AUPRC, evaluated on the test sets. Additionally, the robustness of the models against intra-dataset distribution discrepancies is assessed by comparing delta values, *i.e.*, the absolute difference between the performance on validation and test sets. The generalization ability of the models to unseen tasks is evaluated by conducting few-shot adapting

experiments on a small, unseen, out-of-domain dataset. These experiments are conducted on five random seeds for all models, and the results are averaged across the seeds.

Table 1: **Results on MedTS Cohort for disease classification.** Best results in non-TSM models are highlighted in **bold**, and the best results across all models are underlined. Our model, FORMED, consistently outperforms the other non-TSM model across all datasets on F1 along with many other metrics, and achieves highly competitive performance with SOTA TSM models. The delta values are shown in parentheses: lower delta values indicate more robustness against intra-dataset variances.

| Datasets | Adaptation | Models | Accuracy | Precision | Recall | F1 score | AUROC | AUPRC |
|---|---|---|---|---|---|---|---|---|
| **PTB** (2-Classes) | TSM | Autoformer | 73.35 (17.17) | 72.11 (5.51) | 63.24 (8.45) | 63.69 (7.51) | 78.54 (2.78) | 74.25 (6.62) |
| | | Crossformer | 80.17 (11.51) | 85.04 (9.66) | 71.25 (7.19) | 72.75 (6.92) | 88.55 (3.64) | 87.31 (7.54) |
| | | FEDformer | 76.05 (15.54) | 77.58 (5.72) | 66.10 (8.12) | 67.14 (6.70) | 85.93 (3.01) | 82.59 (7.71) |
| | | Informer | 78.69 (13.96) | 82.87 (5.60) | 69.19 (7.54) | 70.84 (6.07) | 92.09 (1.77) | 90.02 (10.05) |
| | | iTransformer | 83.89 (6.14) | 88.25 (17.43) | 76.39 (3.05) | 79.06 (5.17) | 91.18 (1.80) | 90.93 (19.78) |
| | | MTST | 76.59 (18.40) | 79.88 (6.57) | 66.31 (14.20) | 67.38 (15.61) | 86.86 (4.61) | 83.75 (2.75) |
| | | Nonformer | 78.66 (14.59) | 82.77 (3.94) | 69.12 (9.66) | 70.90 (7.89) | 89.37 (1.22) | 86.67 (5.19) |
| | | PatchTST | 74.74 (20.40) | 76.94 (10.95) | 63.89 (15.42) | 64.36 (18.50) | 88.79 (5.47) | 83.39 (4.65) |
| | | Reformer | 77.96 (14.80) | 81.72 (4.22) | 68.20 (8.55) | 69.65 (7.36) | 91.13 (0.86) | 88.42 (9.28) |
| | | Transformer | 77.37 (15.43) | 81.84 (4.38) | 67.14 (10.22) | 68.47 (8.93) | 90.08 (2.08) | 87.22 (7.22) |
| | TSA | PatchTST-TSA | 78.61 (11.68) | 80.32 (7.87) | 68.74 (2.87) | 70.07 (5.97) | 93.28 (1.51) | 97.15 (1.83) |
| | GA | FORMED (Ours) | 86.24 (3.62) | 89.27 (7.20) | 79.36 (4.18) | 82.11 (4.19) | 95.45 (3.01) | 97.33 (1.08) |
| **PTB-XL** (5-Classes) | TSM | Autoformer | 61.68 (0.87) | 51.60 (2.28) | 49.10 (1.53) | 48.85 (1.75) | 82.04 (0.82) | 51.93 (1.92) |
| | | Crossformer | 73.30 (1.37) | 65.06 (1.60) | 61.23 (1.83) | 62.59 (1.80) | 90.02 (0.66) | 67.43 (1.84) |
| | | FEDformer | 57.20 (0.46) | 52.38 (1.35) | 49.04 (1.27) | 47.89 (1.41) | 82.13 (0.52) | 52.31 (1.44) |
| | | Informer | 71.43 (1.36) | 62.64 (1.76) | 59.12 (2.20) | 60.44 (2.08) | 88.65 (0.81) | 64.76 (2.20) |
| | | iTransformer | 69.28 (0.83) | 59.59 (1.28) | 54.62 (1.58) | 56.20 (1.62) | 86.71 (0.73) | 60.27 (1.79) |
| | | MTST | 72.14 (1.00) | 63.84 (1.40) | 60.01 (1.64) | 61.43 (1.61) | 88.97 (0.64) | 65.83 (2.02) |
| | | Nonformer | 70.56 (1.36) | 61.57 (2.10) | 57.75 (2.33) | 59.10 (2.26) | 88.32 (0.94) | 63.40 (2.52) |
| | | PatchTST | 73.23 (1.07) | 65.70 (1.53) | 60.82 (1.90) | 62.61 (1.86) | 89.74 (0.60) | 67.32 (2.28) |
| | | Reformer | 71.72 (1.09) | 63.12 (1.34) | 59.20 (1.74) | 60.69 (1.60) | 88.80 (0.73) | 64.72 (1.98) |
| | | Transformer | 70.59 (1.25) | 61.57 (1.82) | 57.62 (2.04) | 59.05 (1.96) | 88.21 (0.81) | 63.36 (2.17) |
| | TSA | PatchTST-TSA | 61.45 (0.69) | 53.38 (2.13) | 43.78 (1.43) | 44.41 (1.63) | 82.40 (0.66) | 51.36 (1.62) |
| | GA | FORMED (Ours) | 71.31 (0.79) | 63.94 (1.87) | 56.40 (1.47) | 57.58 (1.77) | 88.44 (0.92) | 63.67 (2.65) |
| **TDBrain** (2-Classes) | TSM | Autoformer | 87.33 (7.23) | 88.06 (6.72) | 87.33 (7.23) | 87.26 (7.29) | 93.81 (4.96) | 93.32 (5.42) |
| | | Crossformer | 81.56 (12.81) | 81.97 (12.47) | 81.56 (12.81) | 81.50 (12.87) | 91.20 (7.38) | 91.51 (7.08) |
| | | FEDformer | 78.13 (16.85) | 78.52 (16.56) | 78.13 (16.85) | 78.04 (16.93) | 86.56 (12.43) | 86.48 (12.51) |
| | | Informer | 89.02 (5.79) | 89.42 (5.66) | 89.02 (5.79) | 88.98 (5.82) | 96.64 (2.66) | 96.75 (2.57) |
| | | iTransformer | 74.67 (12.00) | 74.71 (12.07) | 74.67 (12.00) | 74.65 (12.00) | 83.37 (10.02) | 83.73 (9.60) |
| | | MTST | 76.96 (13.65) | 77.24 (14.51) | 76.96 (13.65) | 76.88 (13.65) | 85.27 (12.28) | 82.81 (13.93) |
| | | Nonformer | 87.88 (8.02) | 88.86 (7.16) | 87.88 (8.02) | 87.78 (8.11) | 97.05 (2.31) | 96.99 (2.35) |
| | | PatchTST | 79.25 (11.04) | 79.60 (11.82) | 79.25 (11.04) | 79.20 (11.01) | 87.95 (9.92) | 86.36 (11.10) |
| | | Reformer | 87.92 (7.02) | 88.64 (6.46) | 87.92 (7.02) | 87.85 (7.08) | 96.30 (2.92) | 96.40 (2.84) |
| | | Transformer | 87.17 (7.85) | 87.99 (7.19) | 87.17 (7.85) | 87.10 (7.92) | 96.28 (2.82) | 96.34 (2.74) |
| | TSA | PatchTST-TSA | 75.50 (13.50) | 77.23 (12.45) | 75.50 (13.50) | 75.09 (13.86) | 82.28 (14.48) | 84.73 (12.19) |
| | GA | FORMED (Ours) | 89.56 (3.42) | 89.94 (3.66) | 89.56 (3.42) | 89.53 (3.44) | 96.25 (2.84) | 96.89 (2.06) |
| **APAVA** (2-Classes) | TSM | Autoformer | 68.64 (7.87) | 68.48 (8.33) | 68.77 (8.69) | 68.06 (8.20) | 75.94 (11.64) | 74.38 (11.74) |
| | | Crossformer | 73.77 (8.12) | 79.29 (6.07) | 68.86 (10.40) | 68.93 (11.18) | 72.39 (20.13) | 72.05 (19.55) |
| | | FEDformer | 74.94 (10.26) | 74.59 (8.07) | 73.56 (7.11) | 73.51 (8.89) | 83.72 (15.66) | 82.94 (17.12) |
| | | Informer | 73.11 (5.18) | 75.17 (5.93) | 69.17 (5.99) | 69.47 (6.49) | 70.46 (14.33) | 70.75 (14.59) |
| | | iTransformer | 74.55 (9.03) | 74.78 (8.49) | 71.76 (11.37) | 72.30 (10.78) | 85.59 (6.15) | 84.39 (6.90) |
| | | MTST | 71.14 (15.51) | 79.30 (8.65) | 65.27 (19.58) | 64.01 (21.71) | 68.87 (25.53) | 71.06 (22.50) |
| | | Nonformer | 71.89 (5.29) | 71.80 (5.85) | 69.44 (5.86) | 69.74 (5.96) | 70.55 (15.03) | 70.78 (14.44) |
| | | PatchTST | 67.03 (15.93) | 78.76 (6.74) | 59.91 (20.38) | 55.97 (25.34) | 65.65 (27.19) | 67.99 (24.14) |
| | | Reformer | 78.70 (2.33) | 82.50 (2.82) | 75.00 (3.19) | 75.93 (3.16) | 73.94 (14.21) | 73.94 (14.21) |
| | | Transformer | 76.30 (3.03) | 77.64 (3.67) | 73.09 (3.31) | 73.75 (3.55) | 72.50 (13.18) | 73.23 (12.77) |
| | TSA | PatchTST-TSA | 69.80 (4.71) | 79.62 (13.96) | 63.49 (6.55) | 61.25 (7.41) | 74.78 (8.71) | 74.36 (12.24) |
| | GA | FORMED (Ours) | 76.46 (8.84) | 77.11 (8.08) | 74.42 (11.68) | 74.65 (10.50) | 82.13 (11.86) | 83.69 (12.40) |
| **ADFTD** (3-Classes) | TSM | Autoformer | 45.25 (4.53) | 43.66 (5.28) | 42.96 (6.02) | 42.59 (4.96) | 61.02 (4.80) | 43.10 (5.42) |
| | | Crossformer | 50.45 (7.53) | 45.57 (11.71) | 45.88 (11.27) | 45.50 (11.51) | 66.45 (4.77) | 48.33 (6.26) |
| | | FEDformer | 46.30 (4.79) | 46.05 (4.52) | 44.22 (5.82) | 43.91 (4.52) | 62.62 (4.98) | 46.11 (5.41) |
| | | Informer | 48.45 (5.12) | 46.54 (6.95) | 46.06 (6.15) | 45.74 (6.49) | 65.87 (2.26) | 47.60 (4.22) |
| | | iTransformer | 52.60 (2.79) | 46.79 (6.02) | 47.28 (6.30) | 46.80 (5.83) | 67.26 (3.29) | 49.53 (3.93) |
| | | MTST | 45.60 (3.30) | 44.70 (2.73) | 45.05 (2.65) | 44.31 (2.60) | 62.50 (2.36) | 45.16 (2.27) |
| | | Nonformer | 49.95 (2.87) | 47.71 (5.54) | 47.46 (4.39) | 46.96 (4.66) | 66.23 (2.22) | 47.33 (5.89) |
| | | PatchTST | 44.37 (7.57) | 42.40 (7.97) | 42.06 (8.24) | 41.97 (7.12) | 60.08 (8.03) | 42.49 (8.78) |
| | | Reformer | 50.78 (2.18) | 49.64 (4.08) | 49.89 (2.30) | 47.94 (2.62) | 69.17 (2.03) | 51.73 (3.93) |
| | | Transformer | 50.47 (3.49) | 49.13 (4.48) | 48.01 (3.84) | 48.09 (3.83) | 67.93 (2.40) | 48.93 (3.92) |
| | TSA | PatchTST-TSA | 50.95 (5.90) | 53.34 (7.91) | 43.50 (1.47) | 40.61 (3.93) | 62.77 (3.56) | 46.89 (5.80) |
| | GA | FORMED (Ours) | 47.76 (2.54) | 46.58 (2.27) | 43.26 (3.92) | 43.05 (2.46) | 61.70 (2.82) | 44.31 (2.46) |

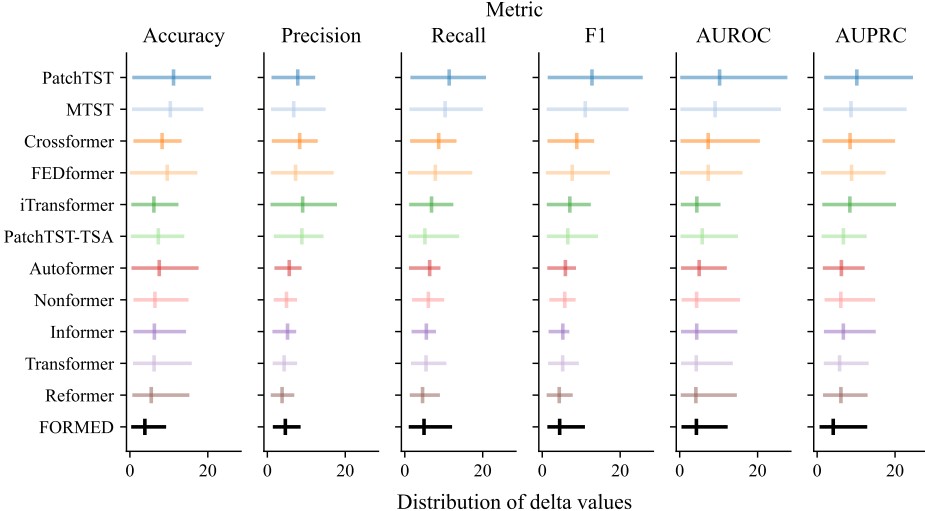

Figure 4: Evaluation of model consistency and robustness across six metrics: accuracy, precision, recall, F1, AUROC, and AUPRC. X-axis: delta values, calculated as the absolute difference between validation and test sets, lower is better; Y-axis: models for comparison, ordered by average delta values. The delta values are collected from 5 datasets for each model and each metric. The range of delta values (minimum and maximum) are indicated by the horizontal lines, and the average delta values are shown with vertical marks. Joint training of multiple datasets helps to reduce the delta values (compare PatchTST-TSA with PatchTST), yet it still falls far behind many other models including ours. Our model consistently exhibits smaller delta values across all metrics, indicating superior robustness and consistency against distributional discrepancies among subjects.

## 5.1 EVALUATION ON REPURPOSING: GENERALIZE TO UNSEEN SUBJECTS

**Setup**. For repurposing datasets in MedTS cohort, we trained 50 TSM models (10 models for each), and 1 TSA model but with 5 task-specific heads. our one FORMED model is trained on all 5 datasets with no change to it during repurposing.

**Effectiveness of Repurposing**. We find that repurposing with a generalizable adaptation layer is more effective than TSM and TSA methods in classification tasks. As shown in Table 1, our model surpasses the TSA model in F1 across all datasets, as well as many other metrics. On top of that, it achieves competitive performance compared to the TSM models, if not better, on most datasets. These findings demonstrate the overall effectiveness of our proposed repurposing framework.

**Quality of Repurposing**. The repurposing also grants the model more robustness towards intra-dataset discrepancies across subjects. The delta values of our repurposed model across six key metrics Figure 4 outperform all 11 baselines, showcasing its consistency and robustness against such variations in data. This implies the applicability of our methods towards real-world healthcare usage, where the subject population at the time of testing is often not fully represented in the training data.

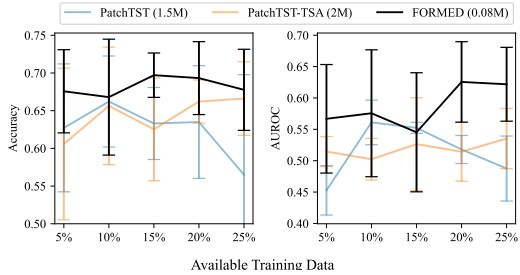

Figure 5: Performance on few-shot adapting to small, unseen, out-of-domain dataset. Numbers of trainable parameters are included in parenthesis. The performance is plotted against the ratio of available training data. FORMED dominates other models across all data ratios in both metrics.

## 5.2 EVALUATION ON
ADAPTING: GENERALIZE TO UNSEEN TASK

**Setup**. In few-shot adapting evaluation, we use a small out-of-domain dataset with a limited amount of training data and a binary classification task. The data is recordings of phonocardiogram (PCG), and is pre-processed into spectrogram time series, where 61 channels each represent a different frequency band. The PatchTST-TSA previously trained on MedTS cohort is modified with a new head, and the PatchTST with reduced parameters is also included for a fair comparison of

both the PatchTST-TSA and our model. Our repurposed model is frozen, and only the newly added channel embedding and task query are learnable.

**Results**. We find that adapting to even drastically different dataset and different task is easily achievable with our model. Despite the such inter-dataset heterogeneity, our model outperforms all baselines across all data ratios in both accuracy and AUROC Figure 5. Interestingly, the PatchTST's performance drops with more available data, and a potential explanation to it is that it quickly memorizes the few training data and comes to an early stop. Nonetheless, our method demonstrates the superior generalization ability to unseen tasks, a significant advantage for use in real-world healthcare applications, where new tasks may arise frequently, and the expert-labeled data is often limited.

## 6 CONCLUSION AND DISCUSSION

In this paper, we present FORMED, a foundation model for MedTS classification, that leverages a pre-trained backbone that can capture general time series patterns and a generalizable adaptation head to repurpose the model and capture domain-specific knowledge. We demonstrate that FORMED can effectively generalize both within and across datasets, providing superior performance with more robustness against distribution discrepancies compared to state-of-the-art models, and can be seamlessly adapted to unseen MedTS datasets with lightweight training. Next, we discuss the potential impact of our work, the limitations, and future directions.

**Potential Impact**. Our work has mainly focused on field of MedTS classification, where leakage of patient information and bias in the model are critical concerns. Regarding the former, we only use datasets that are publicly available and have been de-identified, and the details and sources of them are provided in Table 3. As for the latter, we have taken steps to ensure that our model is fair, such as using a backbone model that has been pre-trained on the largest dataset to capture more general time series patterns, and no covariate information is used other than dataset-level embeddings. Yet, we acknowledge that there may still be biases in the data that we have not accounted for, and we are to release the weights of our model along with a detailed model card (Mitchell et al., 2019) for our community to assess the potential bias and privacy concerns in a joint effort.

**Relation to TimesFM**. The TimesFM (Das et al., 2024) is a foundation model whose sole purpose is time series forecasting, and by repurposing it, we create FORMED which is now a foundation model for medical time series classification, fundamentally different from TimesFM.

**Backbone and Repurposing Domain**. The proposed repurposing-and-adapting framework is not limited to specific backbone model, and can be applied to other time series foundation models with similar anatomy. Our framework can repurpose to other domains like weather forecasting, financial tasks, etc., by simply using a domain-specific repurposing data cohort.

**Computation Efficiency**. Computational cost is of great concern for large foundation models, especially so when the model needs to be frequently adapted to new downstream tasks (Hu et al., 2021). We have recognized such need and already incorporated several strategies to make our model more computationally efficient. By freezing the backbone model and omitting pre-backbone adapters, the gradients do not need to be back-propagated through the backbone model during repurposing (Figure 1), which significantly reduces the computational cost. Moreover, we take what is categorized as an *external memorization* approach (Wang et al., 2024b), where new knowledge of specific tasks is stored in the task-specific embeddings and queries, rather than tuning the model parameters, which further reduces the computational cost at the adaptation stage. On the whole, our model is designed to be computationally efficient and scalable to larger datasets and more complex tasks.

**Interpretability and Explainability**. When it comes to medical applications, interpretability and explainability are crucial for the model to be trusted and adopted by healthcare professionals. As our model is fully transformer-based, it can harness the power of tools that dissect the attention mechanism like Chefer et al. (2021); Hao et al. (2021). Moreover, the task-specific knowledge is explicitly stored in channel embeddings and label queries, which can be used to compare and explain the model's behavior across different tasks. However, all these are beyond the scope of this paper and deserves to be explored in future work.

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

## A    COMPARISON OF ADAPTATION TECHNIQUES

As discussed in Section 2, adaptation techniques for foundation models mainly includes *Prompting*, *Fine-tuning*, *Re-programming*, and *Re-purposing*. We have introduced re-programming and re-purposing, and here we provide a brief overview of prompting and fine-tuning, and compare these techniques based on three aspects: *Data Efficiency*, *New Task Type*, and *Generalizability*.

*Prompting & Fine-tuning*: Both are common adaptation techniques for foundation models, where prompting involves conditioning the model with specific instructions or cues, either handcrafted (Zhou et al., 2023b; Reynolds & McDonell, 2021) or learned through data (Zhou et al., 2022a), and fine-tuning involves updating the model's internal parameters on dedicated dataset (Howard & Ruder, 2018; Ding et al., 2023). While they focus on different aspects of adaptation, they share the commonality of not altering the model's core architecture, therefore the functionality of the model remains unchanged, e.g., model for forecasting remains a forecasting model. Moreover, fine-tuning is often more data-greedy, as it requires updating the whole model's parameters, while prompting only requires learning a few task-specific embeddings or prompts.

In general, these techniques can be categorized based on three aspects: *Data efficiency*, as the scale of dataset used for adaptation, typically measured by the number of parameters updated; *New Task Type*, as the ability to adapt to new tasks that are different from the original task, such as from forecasting to classification; and *Generalizability*, as the ability for the adapted model to be used on unseen datasets and share knowledge across tasks. Table 2 provides a comparison of these techniques based on these aspects.

Table 2: Comparison of adaptation techniques of time series foundation models.

| Adaptation | Data Efficiency | New Task Type | Generalizability |
|---|---|---|---|
| Prompting | ✓ | | ✓[1] |
| Fine-tuning | | | ✓ |
| Re-programming | | ✓ | |
| Re-purposing | ✓ | ✓ | ✓ |

## B    DATA AVAILABILITY

Here we provide the details of the datasets Table 3 used as the MedTS cohort for repurposing in Section 5. The datasets are publicly available, and we follow the pre-processing and splitting procedures as in Wang et al. (2024c).

Table 3: **MedTS Cohort Datasets**.

| Dataset | Type | # Subject | # Sample | Sampling Rate | Sampling Length | # Channel | # Classes |
|---|---|---|---|---|---|---|---|
| PTB (Goldberger et al., 2000) | ECG | 198 | 64 356 | 250 Hz | 300 | 15 | 2 |
| PTB-XL (Wagner et al., 2020) | ECG | 17 596 | 191 400 | 250 Hz | 250 | 12 | 5 |
| TDBrain (van Dijk et al., 2022) | EEG | 72 | 6240 | 256 Hz | 256 | 33 | 2 |
| APAVA (Escudero et al., 2006) | EEG | 23 | 5967 | 256 Hz | 256 | 16 | 2 |
| ADFTD (Miltiadous et al., 2023; ADF) | EEG | 88 | 69 762 | 256 Hz | 256 | 19 | 3 |

---

[1]Although the model structure is fixed and still applicable to other datasets and tasks, the engineered or learned prompts can be task-specific.

