# OpenReview forum: "Repurposing Foundation Model for Generalizable Medical Time Series Classification"
_ICLR.cc/2025/Conference — ICLR 2025 Conference Withdrawn Submission_

### Official Review · Reviewer_m8GH · 2024-10-26

**Soundness:** 2
**Presentation:** 2
**Contribution:** 2
**Rating:** 3
**Confidence:** 4

**Summary:**

This paper introduces FORMED, a foundation model repurposed for MedTS classification. FORMED takes "re-purposing" to integrate a pre-trained backbone model with the medical domain knowledge learned from a curated cohort of MedTS datasets . And "re-purposing" allows the FORMED to adapt to diverse datasets with varying channels, lengths, and tasks. The results show that  FORMED achieves competitive or superior performance, compared with 11 specifically trained baselines.

**Strengths:**

1. Generalizability: The particular "repurposing" enables the FORMED to seamlessly different data characteristics and tasks, which is important  for the MedTS
2. Efficiency: Fine-tuning under the "repurposing" requires minimal parameter updates,  when adapting to new datasets.
3. Performance: Experimental results show that FORMED outperforms or matches the specific-trained baseline models in MedTS classification tasks.

**Weaknesses:**

1. Novelty: The technique novelty has not been clarified in this paper: (1) The keyword "repurposing" is not explained in the Abstract section, and is only merely mentioned with "a specialized shell enriched with medical knowledge. " in the Introduction section. However, How specialized the shell is, which makes it integrate domain knowledge from diverse datasets and diagnosis tasks, has not been emphasized. (2) "curated cohort of MedTS data" has been mentioned in many times in both the Abstract and Introduction. It seems that the collection of the curated cohorts is more important than the design in "repurposing".

2. Experiment: More experiments are required to prove the effectiveness of the proposed method (1) The authors emphasize that "repurposing" enables minimal modification & lightweight parameter update for a specific task in the adaptation. However, no experiment results prove this point by comparing it with existing task-specific adaption methods.  Meanwhile, the extra "repurposing" phase may introduce more parameter updates, even if the update of the "adapting" phase can be lightweight. (2) This paper takes the TimesFM as the backbone. However, it lacks a comparison with the TSA-TimesFM. (3) Lack of ablation study to indicate the effectiveness of the delicate design in the “repurposing”.

**Questions:**

Seen in the weakness.
My primary questions are about the unclarified novelty and lack of experiment

---

### Official Review · Reviewer_k9DD · 2024-11-02

**Soundness:** 3
**Presentation:** 4
**Contribution:** 3
**Rating:** 6
**Confidence:** 4

**Summary:**

This paper introduces FORMED, a foundation model designed for medical time series (MedTS) classification. FORMED repurposes a pre-trained backbone model, originally created for general time series forecasting, to address key challenges in MedTS, including dataset heterogeneity and limited data availability. By leveraging a generalizable adaptation layer, FORMED adapts effectively to various MedTS datasets, handling differences in channels, sample lengths, and diagnostic tasks. The model demonstrates strong generalization across five datasets, and proves versatile in adapting to new datasets with minimal parameter updates.

**Strengths:**

This is a well-written and well-structured paper that proposes a robust foundation model, FORMED, for Medical Time Series (MedTS) classification. FORMED introduces a new approach to repurposing a foundation model for MedTS, tackling challenges of dataset heterogeneity and adaptability in a novel way. By freezing the backbone model and only adapting task-specific layers, it achieves high computational efficiency, allowing it to be fine-tuned on new datasets with minimal data. The model is thoroughly tested on five MedTS datasets and compared to multiple baselines across key metrics,  demonstrating its robustness and consistency across diverse domains.  Overall, this paper offers a meaningful contribution, establishing a strong foundation for future work in MedTS classification.

**Weaknesses:**

1. Although five datasets are used to test generalizability, some datasets are quite similar, such as PTB and PTB-XL or datasets with Alzheimer’s data differing only by channel configurations. Testing on a wider range of datasets with different tasks could better showcase FORMED’s adaptability and robustness.

2. Most baseline models are Transformer-based; including comparisons with ResNet-based models, especially those tailored for MedTS tasks (like 12-lead ECG classifiers achieving strong results on PTB-XL), could provide a more comprehensive evaluation of FORMED’s performance.

3. The paper could benefit from additional experiments to deepen the analysis. For example, evaluating how FORMED performs when trained on a single dataset rather than a diverse set would provide insight into how training on multiple datasets impacts generalization. Such experiments could reveal how the model leverages knowledge across datasets and whether it shows improved performance or robustness compared to training on a single dataset alone.

4. Although Medformer is mentioned in the paper, it is not included as a baseline model. Adding Medformer to the baseline comparisons would provide a more comprehensive evaluation and better highlight FORMED's performance relative to existing MedTS models.

**Questions:**

1. Given that several datasets used are quite similar, have you considered using datasets from distinctly different tasks to better evaluate FORMED’s adaptability?

2. Why did you choose primarily Transformer-based TSM models as baselines, and would you consider comparing FORMED to another architecture such ResNet-based models?

3. Have you conducted more experiments?  how FORMED performs when trained on a single dataset alone, and if not, would you consider this for a better understanding of dataset influence?

4. A minor typo: The terms "label queries" (LQs) and "task query" are used inconsistently in the paper, creating some confusion. Ensuring consistency in terminology would improve clarity.

---

### Official Review · Reviewer_MrEL · 2024-11-02

**Soundness:** 2
**Presentation:** 2
**Contribution:** 3
**Rating:** 3
**Confidence:** 4

**Summary:**

In this paper, the authors present the FORMED architecture, which allows:
- The repurposing of a large time series foundational model from a forecasting task to a classification task;
- The adaptation of that repurposed model on new datasets, with potentially different numbers of channels and target classes.

This is achieved through the use of trainable channel and label embeddings, while the backbone remains frozen. A combination of a Transformer decoder layer and a residual network is also employed to form the desired output: it is trained during repurposing but frozen during adaptation.

FORMED is thus a lightweight model: training it after repurposing only requires training of the channel and task embeddings.

The authors use a pretrained TimesFM backbone and demonstrate the high performance of their repurposing approach on five medical datasets, and of their adapting approach on fractions of a sixth unseen medical dataset.

**Strengths:**

**Originality :** As remarked upon by the authors, research efforts on medical TS often involve task-specific models, or in the best cases task heads that need to be trained for each task. Moreover, when the input dimensions change (apart from the sequence length, whose variation is well handled by the Transformer architecture), input adapters also need to be trained. The authors' approach is original in the sense that they do away with both of those requirements by placing the training-required modules after the weight-heavy backbone.

**Quality :** The authors have tested their repurposing method extensively, providing comparisons with 11 different models on 5 different datasets.

**Clarity :** The provided figures are clear and convey information in a very efficient and readable way.

**Significance :** The presented technique has the potential to greatly reduce the cost (time- and data-wise) of training a model for a new dataset. This is significant, as medical institutions may lack computational resources, large enough datasets, and/or the time to train a new model from scratch.

**Weaknesses:**

This paper suffers from one critical weakness: as the performance of FORMED is in most cases very close or below that of a task-specific model, any motivation behind its use lands entirely on its alleged lightness. Unfortunately, the authors do not provide any quantitative results demonstrating the time and computational efficiencies of their method, especially when compared to task-specific and task-adapted models; they merely provide arbitrary parameter counts.

Moreover, the most common scenario for medical institutions is supposed to be adaptation of a repurposed model, as it requires the least amount of data and computational resources. Yet, the proof provided by the authors that adaptation works is very limited (fractions of a single dataset, only two competing models, and again no efficiency results).

Finally, the claims at the end of section 4.3 are scarcely substantiated: for example, the "domain knowledge" gained by the SDA during repurposing is never demonstrated.

In the current state of the paper, it is impossible to assess how significant the authors' contribution is.

Additionally:
- At line 284, the authors state that they are "processing each channel of input individually", which means that before the SDA block and thus within the backbone model where most of the weights are, channels are processed independently. This has been repeatedly reported to hinder performance on EEG and ECG data, and is even mentioned as a drawback of other models by the authors at lines 120-127.
- The bold highlight in Table 1 is unnecessary as it only denotes the best of two models. One highlighting method would be enough.
- The paper should be checked carefully for spelling and grammar errors (see below).

___

Miscellaneous non-exhaustive paper improvement remarks:
- Line 47, "using a pre-trained and fixed backbone foundation models", 'models' should be singular.
- Line 123/124, "in agree with Tan et al. (2024)" should read "as mentioned by Tan et al." or any correct equivalent wording.
- Line 127, "time series data and trained on multiple": a verb such as 'was' is missing before 'trained'.
- Line 131, "not suitable" can be replaced with "unsuitable".
- Line 142, "able to handle new domain of data": 'a' is missing between 'handle' and 'new'.
- Line 143/144, "on its dark side" should be more formal: 'However' is a valid alternative in this context for example.
- Line 147/148, "modification" should most likely be plural; "specific to certain task" is missing an 'a' between 'to' and 'certain'.
- Line 185/186, "The backbone foundation model is frozen in pre-training while trainable in repurposing and adapting": shouldn't this be the opposite?
- Line 284, a 'the' should be added between 'of' and 'input'.
- Line 295/296, There should be an 's' at the end of 'input', a 'a' before 'dynamic number of output classes', a 'a' before 'new dataset' and a 'the' before 'risk of overfitting'.
- Line 320, "all the parameters in SDA is independent on" should read "all the parameters in SDA are independent of".
- Line 326, "the weights in SDA is randomly", 'is' should be 'are'.
- Line 490, "despite the such inter-dataset", either 'the' or 'such' should be removed.
- Line 505/506, "on field of MedTS", a 'the' should be added between 'on' and 'field'.
- Line 537, "deserves", no final 's' is needed here.

**Questions:**

- At line 160/161, you state: "adapting a forecasting model for general classification tasks requires more than simply modifying the prediction layer; it demands a comprehensive redesign and a deeper understanding of the problem space". Would you happen to have a reference on the matter? My understanding was that fine-tuning a task head was in fact quite successful at adapting models to new tasks (from forecasting to classification, from classification to regression, etc), but I would like more information on this subject.
- How do you explain the large delta values that can be observed for some models (for example, 10 points of F1 Score for FORMED on APAVA)? Is there a meaningful difference between your validation and test sets, in size for example?
- For clarification, when adapting FORMED on the PCG dataset, was the SDA trained using all datasets in the MedTS cohort for repurposing or only one of them? If so, which one?

---

### Official Review · Reviewer_BMSM · 2024-11-03

**Soundness:** 3
**Presentation:** 2
**Contribution:** 2
**Rating:** 3
**Confidence:** 3

**Summary:**

The paper introduces FORMED, a new generalization method for foundation models. It introduces a novel mechnanism called re-purposing, designed for generalizable medical time series (MedTS) classification tasks. Generalization in medical time series data is challenging due to inter- and intra-dataset heterogeneity and data insufficiency, which often hinder model adaptability across various datasets. To overcome these issues, FORMED uses a pre-trained foundation model, TimesFM, as a backbone model and employs a two-stage procedure of re-purposing and adapting. In the re-purposing stage,  it learns the weights of the channel embedding, label query, and classifier. During the stage of adapting, it learns only the weights of the channel embedding and label query, while keeping the classifier frozen. This setup allows the channel embedding and label query to be tailored to specific datasets and tasks,  while the classifier retains essential domain knowledge. As a result, FORMED can adapt effectively to new datasets with varying channel configurations, lengths, and diagnostic tasks. Through evaluations on a curated cohort of five MedTS datasets, the model consistently outperforms traditional task-specific and task-adaptive models, maintaining high performance.

I believe this paper aims to introduce a novel transfer learning paradigm specifically for medical time series dataasets. However, I think some modules and configurations are unclear and require more details. Additionally, based on my experience with biosignal data, the performance appears suboptimal compared to existing methods. Furthermore, the model still requires training a very large classifier (8 million parameters), and given its relatively poor performance, I am not seeing clear advantages. If these points can be addressed, I would consider revisiting my assessment.

**Strengths:**

I believe the main strength of this study lies in its clear motivation and the reasonable solutions it proposes.

1. This study clearly addresses the existing challenges, such as inter-dataset heterogeneity, intra-dataset heterogeneity, and data insufficiency. For instance, a biosignal dataset may contain recordings from multiple participants, each with a unique health status, leading to notable intra-dataset heterogeneity. Additionally, across different datasets, biosignals often exhibit distinct patterns; for example, ECG and heart rate data both describe heart function but follow completely different temporal patterns.

2. It clearly identifies the challenges in adapting foundation models for the time-series domain, particularly the need for a dataset-specific alignment module and output layer, which limits the model’s generalization across different datasets.

3. This study proposes a reasonable two-stage solution. In the stage of re-purporsing, it introduces a generalization classifier designed to capture the domain knowledge, through the transformer's attention module. In the adaptation stage, it learns the channel embedding and label query specifically tailored to each dataset and task.

4. In the adaptation stage, it requires learning only 30k parameters, making minimal modifications to the task head and avoiding specificity to a single task.

**Weaknesses:**

The primary weakness of this paper is its suboptimal performance on medical time series datasets compared to existing methods. Additionally, some modules and configurations are not clearly explained. Given that experimental performance is the main concern, I will begin by discussing limitations in the experiments and model comparisons.

1. In the two-stage solution, its re-purposing stage involves a training of large classifier with 8 million parameters. This parameter size is actually larger than that of some pre-trained models on the medical time series data, which raises questions about the benefits of this adaption approach.


    For instance, Cross Reconstruction Transformer (CRT) introduces a dropping-and-reconstruction pre-training paradigm. Its default setup is 6 encoder layers, 2 decoder layers, and an embedding size of 128, resulting in a model with 3.9 million parameters. Both this study and the CRT paper use PTB-XL biosignal data, which is the largest biosignal dataset used in this study. Consequently, a 4-million-parameter Transformer is sufficient for pretraining, while this adaption method needs to train a classifier with 8 million parameters. It is problematic if domain adaptation requires an even larger adapter, as this undermines the idea of efficient adaptation.

    [1] Self-Supervised Time Series Representation Learning via Cross Reconstruction Transformer. IEEE NNLS, 2023.

2. The classification performance of this paper actually is not impressive. In terms of PTB dataset, it is a very simple binary classification task. As it is simple, we did not really see recent works about pre-training or adaption make it as benchmark. For example, a simple convolution-based model with thousands of parameters can achieve an accuracy of 95%, while the baseline of RNN and SVM can also achieve over 90% [2]. However, in this paper, the proposed method and Transformer baselines are ranged from 73% to 86%, signficantly lower than exisitng methods.

    [2] ECG Heartbeat Classification: A Deep Transferable Representation. ICHI, 2018.

3. The experimental results on the PTB-XL dataset are also lower than those of existing methods. For instance, this paper references the Biosignal Transformer, which achieves a balanced accuracy of 84.21%, an AUPRC of 92.21%, and an AUROC of 76.59%. Additionally, the CRT model reports an accuracy of 87.81% and an AUROC of 89.22%. In contrast, FORMED achieves a balanced accuracy of 71.31%, an AUPRC of 63.67%, and an AUROC of 88.44%. Other related works on this dataset also generally achieve accuracy scores over 80%.

    [3] BIOT: Biosignal Transformer for Cross-data Learning in the Wild. NeurIPS, 2023.

4. The experiments also do not show the advantages of FORMED compared to Task-Specific Model (TSM). For example, the baseline among TSMs are 2% higher than FORMED. Morever, most of baselines are designed for forecasting, like Informer, Autoformer, Fedformer, which may not be suitable baseline. This paper may include Transformer-based baselines for classification tasks.

5. The datasets used in this study are relatively easy for classification tasks due to their shorter sampling lengths, ranging from 250 to 300. Specifically, this study uses PTB-XL with a sequence length of 250. In contrast, CRT employs a sequence length of 5000, and BIOT uses a sequence length of 2500, making those tasks more challenging.

6. I also have questions about the rationale behind using the foundation model TimesFM. This paper states, "Repurposing the foundation model involves changing the forecasting head to a classification head", and "Foundation models have showcased their capability in capturing general time series patterns, through pre-training on forecasting tasks". Here, the forecasting pre-training task implies using a sequence to predict another sequence, whereas next-token prediction refers to using a historical sequence to predict the next token.


    However, as far as I know, most foundation models rely on next-token prediction rather than forecasting tasks. It should be noted that TimesFM, with its forecasting-based pre-training, is an unusual case. In comparison, next-token prediction is generally better suited for learning generalizable knowledge, often enabling zero-shot or few-shot learning capabilities [4,5].


    As a result, I am uncertain about the rationale for re-purposing by changing the head, as **the primary purpose** of a foundation model is to capture inherent generalizable knowledge.


    [4] Language Models are Unsupervised Multitask Learners.


    [5] Language Models are Few-Shot Learners.


7. Although this study highlights the challenges of domain adaptation in medical time series data, such as inter-dataset heterogeneity, intra-dataset heterogeneity, and data insufficiency, no experiments demonstrate these aspects.

    Aside from the benchmarking results, there is a lack of informative ablation studies, and no experiments are provided to support various claims made in the paper.


8. Few typos, such as line 202 in page 4 (f->g)

**Questions:**

I have a few questions, and I’m unsure whether they represent limitations of the paper or simply my own confusion. Could you please help clarify?

1. Task-Specific Design: Is the proposed method in this paper specific to classification tasks? If I wanted to adapt it to a different type of task, such as regression or anomaly detection, would it require retraining a new layer to replace the very large classifier?


2. Use of Transformer Decoder Layer for Classification: Instead of a simple linear classifier, the paper employs a Transformer decoder layer called Shared Decoding Attention (SDA). Since the SDA mechanism only generates an output embedding, I’m unclear on how a decoder layer or attention module can perform classification directly. Could you clarify the role of this module in classification?

    I assume it applied label query of **Q \in K \times D** and generates output embedding with a shape of ** K \times D**. And how will it make classification?

3. How does this SDA module adapt to different classification tasks? The paper claims that the attention module gains domain knowledge during the re-purposing stage, but I am uncertain how an attention layer trained on a limited biosignal dataset can generalize to unseen data.

    For example, if the model is re-purposed by training this attention layer on ECG and EEG signals, it might learn domain knowledge specific to heart or brain functions. It is reasonably adapted to related signals like heart rate during the adapting stage. However, how would this model handle entirely unrelated signals in unseen datasets (i.e., inter-dataset Heterogeneity), such as body temperature, pulse rate, respiration rate, or blood pressure?

---

### Note · Authors · 2024-11-27

**Comment:**

Dear reviewers,

Thank you for your time and effort in the reviewing process and providing constructive feedback. We sincerely appreciate it and recognize the missing pieces of our paper. As we are to make major revision of the paper, we shall not further take your time and thus withdraw.

**Withdrawal Confirmation:**

I have read and agree with the venue's withdrawal policy on behalf of myself and my co-authors.